# Membrane Lipid Reshaping Underlies Oxidative Stress Sensing by the Mitochondrial Proteins UCP1 and ANT1

**DOI:** 10.3390/antiox11122314

**Published:** 2022-11-23

**Authors:** Olga Jovanović, Ksenia Chekashkina, Sanja Škulj, Kristina Žuna, Mario Vazdar, Pavel V. Bashkirov, Elena E. Pohl

**Affiliations:** 1Institute of Physiology, Pathophysiology and Biophysics, Department of Biomedical Sciences, University of Veterinary Medicine Vienna, 1210 Vienna, Austria; 2Federal Research and Clinical Center of Physical-Chemical Medicine, 119435 Moscow, Russia; 3A.N. Frumkin Institute of Physical Chemistry and Electrochemistry, 119071 Moscow, Russia; 4Department of Chemistry, Faculty of Science, University of Zagreb, 10000 Zagreb, Croatia; 5Department of Mathematics, University of Chemistry and Technology, 16628 Prague, Czech Republic; 6Scientific Research Institute of System Biology and Medicine, 117246 Moscow, Russia

**Keywords:** lipid shape, bending moduli, lateral pressure profile, reactive aldehydes, mitochondrial membrane protein, lipid–protein interaction, stored curvature elastic stress, protonophoric function

## Abstract

Oxidative stress and ROS are important players in the pathogenesis of numerous diseases. In addition to directly altering proteins, ROS also affects lipids with negative intrinsic curvature such as phosphatidylethanolamine (PE), producing PE adducts and lysolipids. The formation of PE adducts potentiates the protonophoric activity of mitochondrial uncoupling proteins, but the molecular mechanism remains unclear. Here, we linked the ROS-mediated change in lipid shape to the mechanical properties of the membrane and the function of uncoupling protein 1 (UCP1) and adenine nucleotide translocase 1 (ANT1). We show that the increase in the protonophoric activity of both proteins occurs due to the decrease in bending modulus in lipid bilayers in the presence of lysophosphatidylcholines (OPC and MPC) and PE adducts. Moreover, MD simulations showed that modified PEs and lysolipids change the lateral pressure profile of the membrane in the same direction and by the similar amplitude, indicating that modified PEs act as lipids with positive intrinsic curvature. Both results indicate that oxidative stress decreases stored curvature elastic stress (SCES) in the lipid bilayer membrane. We demonstrated that UCP1 and ANT1 sense SCES and proposed a novel regulatory mechanism for the function of these proteins. The new findings should draw the attention of the scientific community to this important and unexplored area of redox biochemistry.

## 1. Introduction

Lipid heterogeneity, observed between the membranes of cells and cell organelles, as well as between membrane leaflets and subdomains, is essential for proper cell functioning [1,2,3,4,5]. The most abundant lipids in cell membranes are phospholipids. Variations in their structure lead to different shapes, which is related to their basic property, intrinsic lipid curvature (*C*_0_) [6,7,8]. Phosphatidylglycerol and phosphatidylcholine (PC), which spontaneously assemble into lamellar bilayer structures at physiological concentration of ions in water, have a cylindrical shape and negligible *C*_0_ (≈0). Non-bilayer lipids or conical lipids such as phosphatidylethanolamine (PE; *C*_0_ < 0) and lysolipids (lysoPC; *C*_0_ > 0) form highly curved lipid structures such as HII phase (PE) or micelles. PE accounts for approximately 25% of all membrane lipids that significantly affect the mechanical properties of membranes [9,10].

The embedding of conical lipids into a flat monolayer composed of cylindrical lipids modifies the lateral pressure profile (LPP) so that the bending moment produced by LPP in the monolayer is no longer zero [11]. The bending rigidity, quantified by the bending modulus, *k*, is an intrinsic mechanical material constant that defines the amount of energy required to deform a membrane out of the spontaneous state [12]. Membrane bending rigidity is critical for membrane remodeling during vesicular transport, exocytosis, and endocytosis. Maintaining the flatness of the lipid monolayer containing conical lipids leads to the induction of the stored elastic curvature stress (SCES) in the bilayer membrane [13]. SCES induced by PE supports the docking and insertion of membrane-associated/peripheral proteins [10,14,15], membrane budding [16], and membrane remodeling [17,18]. SCES is also suggested to influence the functioning of transmembrane proteins. However, only a few studies have shown this relationship [13,19,20,21,22,23,24].

The role of conical lipids is especially apparent in the inner mitochondrial membrane (IMM), which contains the highest percentage of phospholipids with a negative *C*_0_ (38−45% PE and 15−25% cardiolipin [CL]) [25,26]. Their shape supports the characteristic architecture of the IMM, which is necessary for the proper functioning of the mitochondrion [27,28,29,30]. Because PE senses membrane curvature, the curvature-driven redistribution leads to its accumulation in strongly concave regions [31,32,33,34]. As an example, CL and PE support adenosine 5-triphosphate (ATP) synthase dimer formation, generating strong membrane curvature on the top of the cristae and thereby potentiating protein activity [35,36]. 

Under oxidative stress conditions, membrane lipids undergo shape transformation. The products of reactive oxygen species (ROS), which include the biologically important reactive aldehydes (RAs) 4-hydroxy-2-nonenal (HNE) and 4-oxo-2-nonenal, covalently modify the PE headgroup to form RA-PE adducts [37,38,39,40]. RA-PE adducts increase the protonophoric activity of mitochondrial uncoupling protein 1 (UCP1), an essential player in non-shivering thermogenesis [37]. Based on results obtained by mass spectrometry and molecular dynamic (MD) simulations, we previously suggested that the formation of RA-PE adducts reshapes PE, thereby changing the intrinsic curvature of this lipid [41]. Another contribution to the transformation of the lipid shape during oxidative stress in mitochondria comes from the amplified action of membrane-bound phospholipase A2 (mPLA2) [42,43], which converts membrane lipids to lysolipids (*C*_0_ > 0) and free fatty acids (FAs). The released FAs represent the raw material for the production of more RAs, which then target the headgroup of the PE and form more RA-PE adducts. As a result of these events, the ratio of lipids with opposite shapes in IMM can rapidly alter, leading to a modification of the LPP and the bending rigidity of the IMM [44]. 

We hypothesized that the SCES change caused by the lipid shape transformation is a crucial event by which IMM proteins (particularly uncoupling proteins) respond rapidly to oxidative stress. To investigate the impact of ROS-modified lipids on mitochondrial protein function, we combined experimental measurements of membrane elastic properties and total conductance of membranes reconstituted with recombinant UCP1 and ANT1 with calculations of membrane LPPs using MD simulations.

## 2. Materials and Methods

### 2.1. Chemicals

1,2-dioleoyl-sn-glycero-3-phosphocholine (DOPC), 1,2-dioleoyl-sn-glycero-3-phosphoethanolamine (DOPE), 1-oleoyl-2-hydroxy-sn-glycero-3-phosphocholine (OPC), bovine heart cardiolipin (CL), arachidonic acid (AA), hexane, hexadecane, KCl, HEPES, EDTA, Na_2_SO_4_, MES, Tris, adenosine 5-triphosphate (ATP), and guanosine 5-triphosphate (GTP) were purchased from Sigma-Aldrich (Munich, Germany). MPC (1-myristoyl-2-hydroxy-sn-glycero-3-phosphocholine) was obtained from Avanti Polar Lipids (Alabaster, AL, USA). Plain silica beads (22-μm diameter) were from Microspheres-Nanospheres (Corpuscular Co., Cold Spring, NY, USA). Chloroform was purchased from Carl Roth (Karlsruhe, Germany). 4-oxo-trans-2-nonenal (4-ONE) and 4-hydroxy-trans-2-nonenal (4-HNE) were purchased from Cayman Chemical (Ann Arbor, MI, USA) or synthesized as previously described [45].

### 2.2. Reconstitution of UCP1 and ANT1 in Liposomes

Recombinant uncoupling protein (murine UCP1) and adenine nucleotide translocase 1 (murine ANT1) were purified from *E. coli* inclusion bodies and reconstituted into liposomes made of DOPC, DOPE and CL as previously described [46,47]. AA (20:4, ω6), at a concentration of 15 mol%, and lyso-PCs (OPC and MPC), at the concentrations indicated in the figure descriptions, were added to the lipid phase before membrane formation.

### 2.3. Formation of the Planar Lipid Bilayer Membranes and Measurements of the Membrane Electrical Parameters

Solvent-depleted planar lipid bilayers for the experiments with the recombinant proteins UCP1 and ANT1 were formed from proteoliposomes at the tip of a disposable plastic pipette [48]. The disposable container was filled with 0.75 mL buffer containing 50 mM Na_2_SO_4_, 10 mM MES, 10 mM Tris, and 0.6 mM EGTA at pH 7.32 and T = 305 K. Membrane formation and bilayer quality were monitored by capacitance measurements. Current–voltage (*I*–*V*) characteristics were measured by a patch-clamp amplifier (EPC 10; HEKA Elektronik Dr Schulze GmbH, Lambrecht, Germany). Total membrane conductance *G_m_* was calculated from a linear fit of experimental data (*I*) at applied voltages (*V*) in the range of −50 mV to 50 mV [49]. The relative conductance, *G_rel_*, was calculated according to Equation (1): (1)$$G_{rel}=\frac{G_{m}-G_{0}}{G_{0}-G_{AA}}$$
where *G_m_* and *G_0_* are the total membrane conductance of the lipid membranes reconstituted with transmembrane protein and arachidonic acid (*AA*), with or without lysolipids, respectively, while *G_AA_* is the total membrane conductance of the lipid membranes reconstituted with AA alone.

### 2.4. Measurements of the Membrane Elastic Parameters 

Bilayer lipid membranes (BLMs) for the experiments shown in Figure 1 were formed by applying the “painting” technique to mesh openings [50]. In brief, the openings were pretreated with a small drop of lipid mixture (10 mg/mL total lipid) dissolved in decane:octane (1:1 *v*/*v*). The solvents were evaporated under a stream of argon. Then, a small amount of the lipid mixture dissolved in squalene (20–30 mg/mL, total lipid) was painted on the openings of the grid fixed at a small distance from the bottom of a Petri dish filled with a buffer solution. BLMs formed spontaneously after the excess lipid solution was forced to the periphery, forming a “toroidal meniscus”—a reservoir that maintains the lateral tension of the lipid bilayer (Figure 1A). BLMs were made of DOPC, DOPE, CL, and OPC in the ratios indicated in the figures. The experimental chamber was filled with buffer containing 100 mM KCl, 10 mM HEPES, and 1 mM EDTA at pH 7 and room temperature (295 K). We added RAs to the buffer solution, which surrounded the formed BLM in the container and incubated for 15 min, then the nanotubes were pooled. 

Nanotubes (*NT*s) were pulled from BLMs vertically using a fire-polished borosilicate patch pipette with a tip diameter of ~1 μm filled with the same buffer as the experimental chamber. The tip of the pipette was placed in close contact with the BLM. The hydrostatic pressure pulse ruptured the small membrane patch isolated inside the pipette while the pipette rim remained in contact with the parent BLM. The cylindrical *NT* formed spontaneously when the pipette was slowly moved away from the membrane [51]. A precise nanopositioning system (piezo linear actuator and Actuator Controller ESA-CSA, Newport, Irvine, CA, USA) controlled the vertical position of the pipette (Δ*L*). The formation of the *NT* was detected by conductance measurements using Ag/AgCl electrodes placed in the pipette and bath solution. The *NT* radius was recalculated from the ion current *I*, measured by an Axopatch 200 B amplifier (Molecular Devices, Sunnyvale, CA, USA), and acquired by a low-noise data acquisition system (Axon Digidata 1550, Molecular Devices, Sunnyvale, CA, USA). The amplifier was set in the voltage-clamp mode to measure the ion conductance *G* = *I*/*U*. We used the hyperbolic approximation of *G* (Δ*L*) (Figure 1B) to fit the measured conductance dependence [50]. Briefly, the conductance of the *NT* lumen, *G_NT_*, was obtained as the difference between the *G* and the patch leakage conductance, *G_p_* (*G_p_* = *G* at the infinite *NT* length): *G_NT_ = G* − *G_p_*. The vertical asymptotes of the fit provided the length measurement offset, which should be subtracted from Δ*L* to measure the length of *NT (L_NT_)*. *NT* radius (*r_NT_*) was determined as given in Equation (2),
(2)$$r_{NT}=\langle \sqrt{\frac{G_{NT}L_{NT}\rho NT}{\pi }}\rangle$$
where the *ρ_NT_* denotes the specific electrical resistance of the electrolyte inside *NT* lumen. For electrically neutral *NT*, we considered *ρ_NT_* to be equal to the bulk value *ρ_0_*. *ρ_0_* was 1 Ohm*m for the buffer we used (100 mM KCl). For CL-containing membranes the impact of electrical double layer on total ion concentration inside *NT* was taken into consideration [50]. 

Bending modulus, *k*, and lateral tension, *σ*, were measured as previously described [50]. Briefly, the expansion of *NT* under the action of electrical field [52] was used. The same potential difference, *U,* that is applied to the ends of the *NT* to induce an ionic current through its lumen leads to a transmembrane potential in the nanotube wall, which reduces the lateral tension of the nanotube membrane according to Lippmann’s law of electrocapillarity. Since *NT* transmembrane potential grows linearly from 0 at the “BLM” end to *U* at the “pipette’s tip” end of the *NT* (Figure 1A), the expansion of *NT* is non-uniform [52]. However, the hyperbolic dependence of *G* (Δ*L*) of such a non-cylindrical *NT* is preserved, and the effective radius determined from the hyperbolic approximation of *G* (Δ*L*) is related to the elastic characteristics of the membrane and the applied voltage by the Equation (3):(3)$$\frac{1}{{\left(r_{NT}+h\right)}^{2}}=\frac{2\sigma }{k}-\frac{c_{sp}U^{2}}{3k}$$
where *h* = 2 nm is the monolayer thickness and *c_sp_*—the specific capacitance of BLM. 

*c_sp_* ≈ 1 μF·cm^−2^ for the membranes prepared from lipid solution in squalene [53]. Thus, *r_NT_* was measured at different *U* and the dependence ${\left(r_{NT}+h\right)}^{-2}$ on *U*^2^ was plotted. The linear regression of this plot gave *k* as the tangent to the slope of the line and *σ* was determined from the intersection of the line with the ordinate axis (Figure 1C).

### 2.5. Molecular Dynamics Simulations 

We performed molecular dynamics (MD) simulations for lipid bilayers of different compositions—DOPC, DOPC:DOPE (50:50), DOPC:ONE-PE (50:50), and DOPC:OPC (50:50). DOPC, DOPE, ONE-PE Schiff base adduct [37,41], and OPC were described with Slipids force field [54,55,56]. All missing bonding and non-bonding parameters of lipid molecules in the existing Slipids force field were updated with compatible CHARMM36 parameters [57] when needed. Atomic charges were recalculated by the standard Slipids procedure using the Merz–Singh–Kollman scheme [58], which is composed of B3LYP/6-31G (d) geometry optimization of the molecule of interest, a subsequent single point ESP charge calculation using the B3LYP/cc-pVTZ method, and a final charge refinement with the RESP method [59]. 

Bilayers containing 128 lipid molecules were constructed from two monolayers containing 64 individual lipid molecules, with symmetrical lipid distribution across the leaflets in mixed systems. All systems were placed in a unit cell and solvated by about 12,000 water molecules using the TIP3P water model [60]. The unit cell size was approximately 6.5 × 6.5 × 12.0 nm. Three-dimensional periodic boundary conditions were used with long-range electrostatic interactions beyond the non-bonded cut-off of 1 nm using the particle-mesh Ewald procedure [61] with a Fourier spacing of 1.2 nm. Real-space Coulomb interactions were cut off at 1 nm, while van der Waals interactions were cut-off at 1.4 nm. We performed 100 ns MD simulations with semi-isotropic pressure coupling, independently in the directions parallel and perpendicular to the bilayer’s normal using the Parrinello–Rahman algorithm [62]. The pressure was set to 1 bar, and a coupling constant of 10 ps^−1^ was used. All simulations were performed at 310 K and controlled with the Nose–Hoover thermostat [63] independently for the lipid–water sub-systems, with a coupling constant of 0.5 ps^−1^. Bond lengths within the simulated molecules were constrained using the LINCS [64]. Water bond lengths were kept constant by using the SETTLE method [65]. Equations of motion were integrated using the leap-frog algorithm with a time step of 2 fs. For the LPP, a custom version of GROMACS-LS package [66,67] was used to re-run trajectories and output local stress tensors. Because long-range electrostatics via PME is not available in GROMACS-LS, an increased cut-off distance of 2 nm was used for Coulomb interaction calculations as suggested by the package developers [66]. 

### 2.6. Statistics

Data from the electrophysiological measurements are displayed as mean ± standard deviation of at least three technical replicates (on three different days). Each replicate was the mean membrane conductance of three to ten bilayer membranes formed on the same day. Error bars for pressure profiles of lipids were calculated as the difference between symmetrized and unsymmetrized pressure profiles in different leaflets. Smoothed data were obtained from the average of two points.

## 3. Results

### 3.1. Impact of DOPE on Membrane Elastic Parameters and SCES 

First, we examined how the presence of PE influences the elastic properties of lipid bilayer membranes. To do so, we measured the radii (*r_NT_*) of nanotubes (*NT*s) pulled from BLM at different voltage biases (*U*) applied to the *NT* interior (Figure 1A). 

To evaluate the impact of PE on membrane elastic parameters_,_ we compared *NT* pulled from BLM made of either (i) DOPC:DOPE:CL (45:45:10 mol%), mimicking the IMM, or (ii) DOPC:CL (90:10 mol%), as a model of a PE-free membrane. The *r_*NT*_* was measured for both lipid compositions at *U* values varying from 50 to 200 mV. The membrane bending rigidity modulus (*k*) and lateral tension (*σ*) of BLMs were calculated from the linear regression of ${\left(r_{NT}+h\right)}^{-2}\left(U^{2}\right)$ according to Equation (3) (see Section 2).

We found that membranes containing 45 mol% DOPE were considerably less resistant to bending than DOPE-free membranes (Figure 1D). The measured bending modulus decreased from *k* = (22.1 ± 2.0) *k_B_T* for PE-free membranes to *k* = (4.8 ± 1.8) *k_B_T* for PE-containing membranes.

This observation qualitatively confirmed the previously reported effect of membrane softening by PE, demonstrating that the addition of 30 mol% DOPE significantly reduced the *k* [68]. At the molecular level, the effect was attributed to the curvature-induced redistribution of DOPE between *NT* monolayers and the reservoir membrane during *NT* formation. Conical lipids with a pronounced *C*_0_ < 0 tend to accumulate in the inner leaflet and to be deposited from the outer monolayer of the highly curved *NT* membrane, which remained connected to a flat reservoir membrane during the measurement [32,33]. Reduction in the apparent bending rigidity modulus, *k*, due to curvature-driven DOPE redistribution in the membrane can be estimated according to the Equation (4) [69]:(4)$$k=\frac{\tilde{k}}{1+\frac{a\tilde{k}C_{0,DOPE}^{2}\;\varphi \left(1-\varphi \right)}{2k_{B}T}}$$
where $\tilde{k}$ is the bending modulus of membranes with a restricted PE curvature-composition coupling, *a*—the area per lipid (about 0.7 nm^2^), *C*_0,_*_DOPE_*—the intrinsic curvature of DOPE, and *φ*—the molar fraction of DOPE in the lipid reservoir. 

For PE-containing membranes, we considered $\tilde{k}$ equal to the bending modulus measured for PE-free membranes and characteristic of DOPC membranes [50]. Using Equation (4), we calculated *C*_0_,*_DOPE_* ≈ −0.45 nm^−1^, which is in agreement with previously published data obtained for DOPE in the inverted HII phase [70], confirming the reduction in apparent bending rigidity of PE-containing membrane due to PE curvature-composition coupling [44]. At the same time, lateral tension in PE-containing membranes was almost twice as high (*σ* = (1.4 ± 0.4)∙mN∙m^−1^ versus *σ* = (0.8 ± 0.3) mN∙m^−1^ in PE-free membrane). BLM is an open system, in which a lipid bilayer is connected to a large lipid reservoir that defines the chemical potentials of the lipids. The lateral tension, *σ*, is related to the surface free energy, required for keeping lipids in a flat bilayer or the same SCES = $\frac{1}{2}kC_{m}^{2}$*,* where *C_m_* is the average spontaneous curvature of the lipid mixture. Thus, the lateral tension of BLM should increase with DOPE content by the SCES magnitude caused by the insertion of conical PE lipids into a flat membrane. Indeed, we found that *σ* gradually increases in DOPC membranes with increasing DOPE content, so that it almost doubles at 50 mol% DOPE, confirming the direct contribution of SCES to the lateral tension of membranes connected to the lipid reservoir.

We considered the contribution of the CL curvature-composition coupling to the apparent bending rigidity to be the same in both types of the membrane (PE-containing and PE-free) because of its lower concentration (10 mol% of total lipid composition) and smaller *C*_0_ (*C*_0,*CL*_ ≈ −0.15 nm^−1^ [70]) compared to PE.

### 3.2. Reactive Aldehydes Modify the Elastic Properties of the PE-Containing Lipid Bilayer Membranes 

Next, we studied the effects of the modification of the lipids by RAs [37] on the elastic properties of membranes. We incubated the reservoir membranes of different lipid compositions with a RA (4-hydroxy-2-hexenal [HHE], HNE, or ONE) at the concentrations of 0.5–0.7 mM, which are in accordance with the concentrations found under physiological conditions (~0.3 mM) and under oxidative stress (up to 5 mM in cellular membranes) [71,72]. We found that HNE and ONE induced a pronounced reduction in the apparent bending modulus *k* in DOPE-containing membranes, resulting in *k_HNE_* = 11.7 *k_B_T* and *k_ONE_* = 7.1 *k_B_T* (Figure 2A and Appendix A). HHE had no detectable impact on *k* (Figure 2A and Appendix A). We explained the lack of the HHE effect by its weaker adsorption to the lipid membrane [37]. It is noteworthy that in the PE-free membranes, none of the RAs caused a measurable change in the bending modulus *k* (Figure 2B and Appendix A).

Alongside enhanced compliance to bending, we observed a decrease in *σ* for PE-containing membranes after their incubation with RAs (Figure 2A). Again, this effect was absent in PE-free membranes (Figure 2B). MD simulations predicted a significant alteration of the PE molecular shape caused by its modification by RAs [41]. Such lipid reshaping is highly likely to be the reason for the simultaneous reduction in both the *k* and *σ* caused by RAs in PE-containing membranes.

### 3.3. OPC Alters the Elastic Properties of the Lipid Bilayer Membrane 

Next, we measured the impact of inverted conical lipid 1-hydroxy-2-oleoyl-sn-glycero-3-phosphocholine (OPC; *C*_0__,*OPC*_ > 0) on the elasticity of PE-containing membranes. The presence of OPC mimics the accumulation of lysolipids in the IMM due to increased phospholipase activity in response to oxidative stress [42,43]. Therefore, we measured the changes in the *σ* and *k* of DOPC membranes (*C_0,_*_*DOPC*_ ≈ 0) induced by the addition of DOPE alone or simultaneously with OPC in the molar ratios indicated in the figure descriptions (Figure 3 and Appendix A). In these particular studies, we did not add CL to the BLM compositions because we aimed to use lipids whose molecular shapes were as complementary as possible: the absolute *C*_0_ values of DOPE and OPC are practically the same (*C*_0,*OPC*_ = 0.41 nm^−1^ [70]).

Our results showed that the presence of both conical lipids, OPC and DOPE, made membranes significantly more compliant to bending than DOPE alone, and the effect is potentiated with an increase in DOPE by constant OPC. The bending modulus of a DOPC membrane containing DOPC:DOPE:OPC (60:20:20) mol% was *k =* [13.6 ± 0.63] *kT* and lower than that of a DOPC:DOPE (50:50) lipid mixture (*k* = [16.7 ± 1.05] *kT*). The bending modulus of the membrane containing DOPC:DOPE:OPC (30:50:20) mol% was the lowest (*k =* [11.4 ± 0.79] *kT*). Thus, the OPC-induced changes in elastic parameters of PE-containing membranes (Figure 3) were similar to those caused by the accumulation of PE-RAs adducts in the membrane (Figure 2A). Both OPC and PE-RA adducts diminished the SCES initially produced by PE.

The replacement of DOPC with DOPE led to a concentration-dependent increase in the lateral tension *σ* in the lipid bilayer membrane (Figure 3B) [44]. Replacement of 20 mol% DOPC with the lysolipid OPC abolished the DOPE-induced increase in *σ*. This indicates that the effect of OPC on SCES was due to the complementarity of DOPE and OPC shapes (Figure 3B and Appendix A). 

### 3.4. The IMM Proteins UCP1 and ANT1 Sense Stored Curvature Elastic Stress 

To test our hypothesis about the impact of SCES on mitochondrial membrane protein function, we performed measurements of total conductance (*G_m_*) of membranes reconstituted with recombinant UCP1 and ANT1. Both proteins are known to assist FA in the transport of protons from the cytosol to the mitochondrial matrix [73,74,75,76]. To test the role of lipid shape and thus conditioned SCES in the lipid membrane, we reconstituted UCP1 or ANT1 in lipid bilayer membranes containing lysolipids with pronounced positive individual curvature, 1-myristoyl-2-hydroxy-sn-glycero-3-phosphocholine (MPC; 14:0) and OPC (18:1). Because the acyl chain length defines the intrinsic lipid curvature (the shorter the acyl chain, the higher the individual lipid curvature), we achieved different SCESs in the lipid bilayer membrane at the same concentration of lysolipids. 

We compared *G_m_* of membranes formed from (i) DOPE:DOPC:CL (45:45:10 mol%), (ii) MPC:DOPC:DOPE:CL (5:45:40:10 mol%), or (iii) OPC:DOPC:DOPE:CL X:(45 − X/2):(45 − X/2):10, (X = 5, 10, and 12.5 mol%), thereby keeping a constant protein to lipid ratio. In all experiments, the UCP1 and ANT1 activities were measured in the presence of the arachidonic acid (AA; 20:4, ω6), because this fatty acid is released from phospholipids by the activity of phospholipase A2 (PLA2) [77]. With 5 mol% MPC, the *G_m_* (MPC) increased to (124.6 ± 13.7) nS/cm^2^, in contrast to the *G_m_* = (79.9 ± 5.3) nS/cm^2^ in the absence of MPC (Figure 4A). In the experiments with OPC, we demonstrated a concentration-dependent increase in *G_m_*, from *G_m_* = (89.9 ± 6.7) nS/cm^2^ in the absence of OPC to the *G_m_* (OPC) = (158.5 ± 15.8) nS/cm^2^ at 12.5 mol% OPC (Figure 4B). We failed to measure the conductance of the protein-containing membranes at lysolipid concentrations higher than 5 mol% MPC or 12.5 mol% OPC because of the high membrane instability. In the control measurements, we showed that, in the absence of AA, even 15 mol% OPC does not affect UCP1-mediated *G_m_*, which remained comparable to the *G_m_* (OPC) = (11.2 ± 1.5) nS/cm^2^ of neat lipid bilayer membranes (Appendix A). The relative increase in *G_rel, ANT1_* = 1.65 (Equation (1)), mediated by ANT1 reconstituted in lipid bilayer membranes containing 10 mol% OPC (Figure 4C), was comparable to the *G_rel, UCP1_* = 1.5, measured for the membranes reconstituted with UCP1 in the presence of OPC (Figure 4B and Appendix A).

We tested how SCES affects the inhibition of ANT1 by adding 4 mM ATP to the ANT1 reconstituted with 10% OPC (Appendix A). A slight decrease in the relative ANT1 inhibition (~66% in the presence of OPC vs. ~76% in its absence; Appendix A) suggests that the binding of ATP to R79 of ANT1 [74] was unaffected by SCES. 

### 3.5. Impact of Lipid Shape on the Lateral Pressure Profiles across the Lipid Bilayer Membrane

The results shown above suggest that lipids with different intrinsic curvatures promote activation of the membrane proteins to different extents. Therefore, we used MD simulations to investigate the impact of lipid shape on the LPP in lipid membranes comprising (i) DOPC, (ii) DOPC:ONE-DOPE, (iii) DOPC:OPC, and (iv) DOPC:DOPE (Figure 5A–C and Appendix A). We omitted CL from our MD simulations because (1) CL has low intrinsic curvature (*C*_0,*CL*_ ≈ −0.15 nm^−1^) [78], and (2) we focused only on lipids that change shape as a consequence of oxidative stress, such as PE or lysolipids. The ONE-DOPE adduct is chosen as a prototypical Schiff base adduct formed via the reaction of ONE with PE [37], while OPC mimics lysolipids formed due to increased PLA2 activity. The LPP arises in the lipid bilayer membrane due to the repulsive interaction in the lipid headgroup and acyl chain regions and strong attraction at the lipid–water interface.

To highlight the fact that the changes originated from the modified PE headgroup due to the formation of the RA-PE adducts, we compared the LPPs in DOPC:DOPE and DOPC:ONE-DOPE membranes (Figure 5A). A substantial drop in the pressure occurred across the whole membrane profile, in the headgroup region, hydrophobic core, and water–lipid interface. In the membrane–water interface region, *Δp_ONE-PE_* = (*p_PE_* − *p_ONE-PE_*) was equal to 302 bar (25%), while in the acyl chain region, the *Δp_ONE-PE_* was 72.5 bar (40%). To evaluate the contribution of lysolipids to the LPP, we compared the DOPC and DOPC:OPC membrane bilayers (Figure 5B). The decrease in lateral pressure due to the replacement of DOPC with OPC occurred at the water–lipid interface and in the acyl tail regions and was slightly pushed to the center of the bilayer. Compared to the LPP of pure DOPC membranes, the lateral pressure in the OPC-containing membranes (Figure 5B) decreased in the membrane–water interface region, from *Δp_OPC_* = (*p_DOPC_* − *p_OPC_*) = 188.6 bar (15.76%), to *Δp_OPC_* = 182 bar (27%) in the acyl chain region, and to *Δp_OPC_* = 38.6 bar in the center of the bilayer. We also analyzed the effect of DOPE (*C*_0_ < 0) on the LPP in the lipid bilayer by comparing the membrane with DOPC membrane with near-neutral curvature (*C*_0_ ≈ 0). In contrast to OPC and ONE-PE, insertion of DOPE into the lipid bilayer slightly increased the lateral pressure in the acyl chain region (*Δp_DOPE_* = (*p_DOPE_* − *p_DOPC_*) = 256.2 bar [27.5%]), and slightly shifted the profile outwards from the bilayer center (Figure 5C). Importantly, the changes in LPP caused by the introduction of DOPE or ONE-PE or OPC into the DOPC bilayer are in qualitative agreement with the lateral stress measurements (Figure 2 and Figure 3).

We quantified the distribution of LPP for each lipid bilayer by calculating the lateral tension *σ*, which is equal to the area under the pressure profile curve (Appendix A). This allowed us to determine *σ* for the headgroup, water–lipid interface, and hydrophobic regions in the lipid bilayer membrane and compare the results for unmodified and modified lipids (PE vs. ONE-PE, DOPC vs. OPC, and PE vs. PC) (Figure 5D–F). 

Our results showed that transformation of the PE headgroup (PE→ONE-PE) reduced the *σ* value by 29.6%, whereas deletion of the acyl chain (DOPC→OPC)—by 17% (Figure 5D,E). More precisely, the decreases in the *σ* values for the PE→ONE-PE and DOPC→OPC modifications were 10% and 6.5% in the water–lipid interface region and 43.1% and 37.6% in the acyl chains region (Figure 5D,E). In contrast, PE insertion into the lipid bilayer increased the *σ* value by only 4.3% in total, and by 11.9% in the acyl chain (Figure 5F). These observations are in agreement with the data of Zoni et al. [79] that were obtained for LPPs and areas under the curve in the hydrophobic core region for mixed bilayers containing DOPC, DAG, DOPE, and DLPC, where DLPC can be considered a lipid with a positive individual curvature [21]. 

Interestingly, the variability of the lipid acyl chains in the mitochondrion [80] could also influence the modification of the PE headgroup. However, Bacot et al. [39] showed that a reduction in the unsaturation of the mixed acyl chains (18:0/20:4 PE versus 18:0/22:6 PE) only slightly reduced the modification of the PE headgroup by hydroxyalkenals (HHE, HNE, and HDDE).

The results shown in Figure 5D (PE→ONE-PE) are consistent with the experimental results obtained for the PE→ONE-PE modification (Figure 2A). Figure 5E indicates the weakening of the membrane lateral tension due to DOPC→OPC modification but it cannot be directly compared with the result shown in Figure 3B, because the MD simulation did not include DOPE (Figure 5E). As mentioned before, we applied MD simulations exclusively to demonstrate the contribution of lipid shape change to lateral pressure. The very small changes in surface tension, *σ*, of the lipid bilayer due to the presence of DOPE (Figure 5F) are consistent with published data [79,81]. In contrast, an increase of nearly 100% is shown for the same composition of the membrane (Figure 3B). Due to the fundamental difference between the applied experimental and in silico methods, obtained values in Figure 3B and Figure 5F are not directly comparable. A bilayer lipid membrane (Figure 3) is an open system in which a lipid bilayer is connected to a large lipid reservoir. Therefore, lateral tension, *σ*, which arises in the lipid bilayer, is related to the surface free energy density of the lipids from a lipid reservoir. The lateral tension of the membrane increases by the SCES magnitude due to the insertion of the conical PE into the flat membrane. 

In contrast, in the MD simulations (Figure 5), we consider a closed system consisting of a certain number of lipid molecules. In this system, SCES does not directly contribute to the lateral tension, which we calculate as the area under the curve obtained for the lateral pressure distribution. MD simulations in this case show the qualitative change in direction (increase/decrease), but not the quantitative magnitude of the change.

Based on the analysis presented above, the individual curvature of the ONE-PE should be more like that of OPC and, in any case, substantially different from the shape of the endogenous PE. Taken together, our results suggest that membrane protein activity increases if the lateral pressure at the water–lipid interface and in acyl chain regions decreases due to lipid modification; otherwise, it stays unchanged.

## 4. Discussion

The importance of the membrane lipid shape/reshaping for the mitochondria functioning under oxidative stress conditions (Figure 6A) can be summarized in two related categories: (i) softening of the lipid bilayer membrane, and (ii) a regulatory effect on the action of membrane proteins. In our previous study, we demonstrated the increase in *G_m_* by AA-activated UCP1 due to the formation of RA-PE adducts and linked it for the first time to the change in membrane properties (Figure 6B, modified data from [37]).

Here, we show that the integral proteins, UCP1 and ANT1, embedded in IMM-like membranes, sense changes in SCES caused by oxidative stress. The experimental results and MD simulations revealed that the modified phospholipids—PE adducts and lysolipids—induce similar changes in the bending rigidity, lateral tension, and LPP of lipid bilayer membranes (Figure 6C,D). The observed variations in *k* and *σ* among RAs can be explained by differences in the types of RA-PE adducts and their localization in the lipid bilayer membrane [37]. Because OPC (*C*_0__,*OPC*_ > 0) induced similar changes in *k, σ,* and *p* as a ONE-PE adduct in PE-containing membranes, we speculate that this PE adduct has a positive individual curvature. Thus, RA-driven shape transformation in a fraction of PE molecules with a negative *C*_0_ reduced lipid packing stress due to mutual compensation with lipids with the opposite *C*_0_, thereby decreasing the membrane SCES.

More importantly, the impact of PE-adducts on the elastic parameters *k* and *σ* in the order HHE < HNE < ONE perfectly matches their ability to enhance the UCP1-mediated proton translocation in the presence of FAs in membranes of the same lipid composition (Figure 4A) [37]. This supports the hypothesis that RA-induced changes in a lipid environment, and not a modification of protein amino acid residues, increase the *G_m_*. A zwitterionic lysoPC lipids increased UCP1- and ANT1-mediated proton translocation similarly to PE adducts. Thus, we propose that lipid shape and associated membrane mechanical properties, such as bending rigidity and lateral pressure, play a regulatory role in the protonophoric activity of UCPs and ANTs. In contrast, the membrane surface potential, which increases due to the formation of RA-PEs [37], is most likely irrelevant for the activation of UCP1 and ANT1, because zwitterionic lysolipids do not affect the surface potential. 

Notably, MD simulations suggested decreased lateral pressure over the whole membrane profile for bilayer membranes containing ONE-PE adduct or OPC (Figure 6D). We suggest that the altered lipid environment at the molecular level results in increased *G_m_* in the presence of UCP1 or ANT1. The changes in the lipid environment are caused by the nature of the lipid transformation, in which the ratio (lipid head)/(acyl chains) is increased compared to the “initial form” of the lipid (e.g., DOPC→OPC, PE→RA-PE). These modifications decreased both the bending modulus *k* and the lateral pressure *p* in the lipid bilayer membrane (Figure 2A and Figure 5A,B).

The insertion of PE into the PC lipid bilayer resulted in an increase in lateral tension (Figure 3B), while the UCP1-mediated conductance remained unchanged (Figure 4A). In the case of RA-PE and lysolipids, decreased lateral pressure in the acyl tail region (Figure 6D) could increase the probability that anionic free fatty acids (i) reach the protein binding site, which is located in the hydrophobic region on the matrix side of the IMM [74], (ii) are protonated in the position near the center of the bilayer (asterisks in Figure 6D and reference [82]), and dissociate from the protein. Eventually, the same lipid environment supports the protein conformation change, ensuring faster FA^−^ translocation. Notably, protein modifications, such as crosslinking of RA amino acid adducts or protein mutations, can lead to loss of the protein transport function, whereas lipid modification potentiates the protein-mediated FA^−^ translocation. Remarkably, in the absence of oxidative stress lipids with distinctly positive curvature, such as lysolipids and phosphoinositides, are found only in trace amounts (<1%) and are primarily involved in direct lipid–protein interaction and signaling [9].

## 5. Conclusions

Identification of the mechanisms by which the transformation of lipid shape affects the functioning of mitochondria helps to explain the onset of diseases associated with oxidative stress in a way that has not been considered so far. Our results show that lysolipids and PEs modified by RAs similarly affect membrane mechanical properties by decreasing the bending modulus *k* and SCES. Furthermore, we showed that UCP1 and ANT1 sense SCES and proposed a new mechanism for regulating the protonophoric function of IMM proteins under oxidative stress.

## Figures and Tables

**Figure 1 antioxidants-11-02314-f001:**
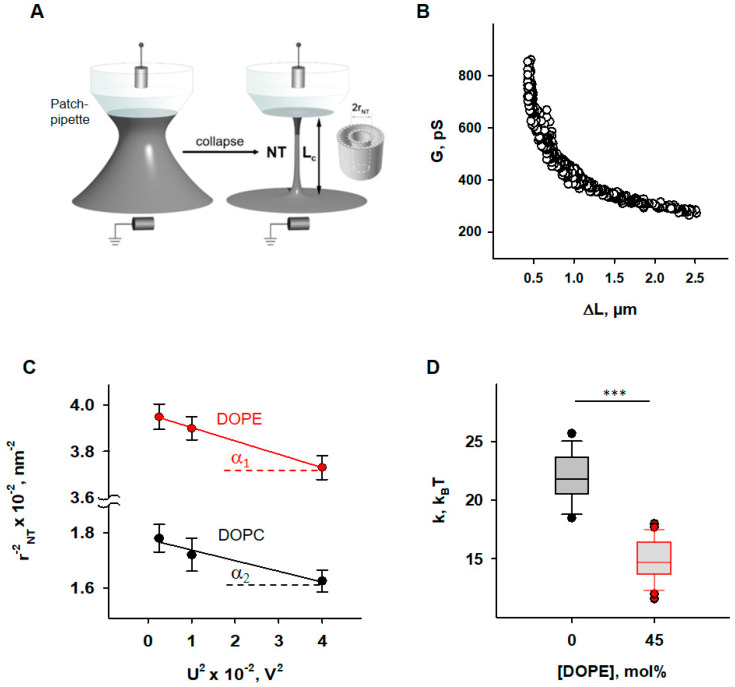
Determination of the elastic parameters of the lipid bilayer membrane based on nanotube (*NT*) pulling. (**A**) Scheme of *NT*s pulled from a bilayer lipid membrane (grey) and held by a patch-pipette (white). A voltage applied to the ends of the *NT*s (U) induces an ion current (*I_NT_*) flowing through the *NT* interior. (**B**) A representative measurement shows the dependence of measured membrane conductance (G) (black circles) on the *NT* length change (ΔL), required for calculation of the *NT* radius (*r_NT_*) (see Section 2). (**C**) Dependence *r_NT_* on U, obtained for the membrane lipid compositions DOPC:CL 90:10 (black) and DOPC:DOPE:CL 45:45:10 (red). (**D**) Box plot and distribution of the bending modulus, *k*, depending on the molar concentration of the phosphatidylethanolamine (DOPE). The buffer solution contained 100 mM KCl, 10 mM HEPES, and 1 mM EDTA at pH 7.0 and T = 295 K. Data points represent the mean and standard deviation from more than 10 independent experiments. *** *p* < 0.001, *t*-test.

**Figure 2 antioxidants-11-02314-f002:**
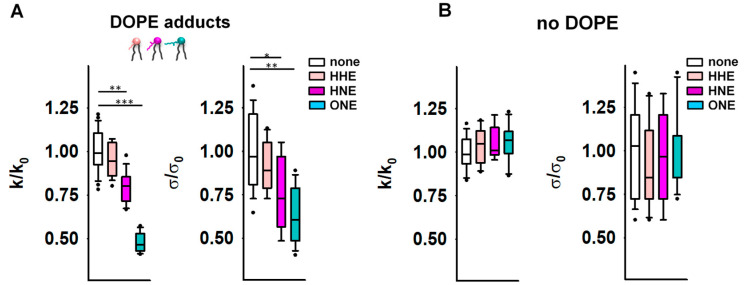
Elastic properties of the lipid bilayer membrane in the presence of reactive aldehydes (RAs). (**A**,**B**) Relative bending rigidity *k/k*_0_ and relative lateral tension *σ*/*σ*_0_ for lipid membranes composed of DOPC:DOPE:CL 45:45:10 in (**A**) and DOPC:CL 90:10 in (**B**), incubated with the RAs 4-hydroxy-2-hexenal (HHE), 4-hydroxy-2-nonenal (HNE), and 4-oxo-2-nonenal (ONE). *k*_0_ and *σ*_0_ are bending modulus and lateral tension in the absence of RAs, respectively. RAs were added in a concentration range of 0.5–0.7 mM. Buffer composition: 100 mM KCl, 10 mM HEPES, 1 mM EDTA pH = 7.0, T = 295 K. Data points represent mean and standard deviation from more than 10 independent experiments. * *p* < 0.05; ** *p* < 0.01; *** *p* < 0.001, *t*-test.

**Figure 3 antioxidants-11-02314-f003:**
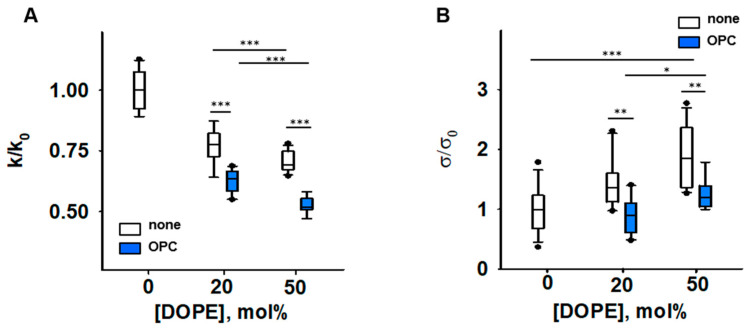
Impact of PE and OPC on the elastic parameters *k/k*_0_ (**A**) and *σ*/*σ*_0_ (**B**). *k*_0_ and *σ*_0_ are bending modulus and lateral tension in the absence of PE, respectively. Buffer solution contained 100 mM KCl, 10 mM HEPES, 1mM EDTA pH = 7.0, T = 295 K. Data points represent mean and standard deviation from more than 10 independent experiments. * *p* < 0.05; ** *p* < 0.01; *** *p* < 0.001, *t*-test.

**Figure 4 antioxidants-11-02314-f004:**
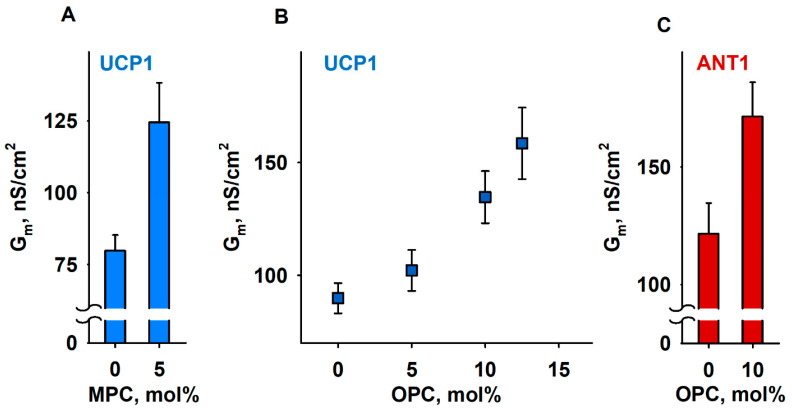
Lipid shape affects UCP1- and ANT1-mediated total membrane conductance (*G_m_*). (**A**,**B**) Dependence of *G_m_* measured in membranes reconstituted with recombinant UCP1 on the concentration of lysolipids MPC (**A**) and OPC (**B**). (**C**). Dependence of *G_m_* measured in membranes reconstituted with recombinant ANT1 on the concentration of OPC. The lipid composition in control experiments was DOPC:DOPE:CL (45:45:10). The specified lysolipid amount (mol%) was used instead of DOPC and DOPE. The concentrations of UCP1 and ANT1 were 4–5 µg/(mg lipid). The concentrations of lipid and AA were 1.5 mg/mL and 15 mol%, respectively. The buffer solution contained 50 mM Na_2_SO_4_, 10 mM MES, 10 mM Tris, and 0.6 mM EGTA, at pH = 7.32 and T = 305 K. Data points represent means and standard deviation from 3–5 independent experiments.

**Figure 5 antioxidants-11-02314-f005:**
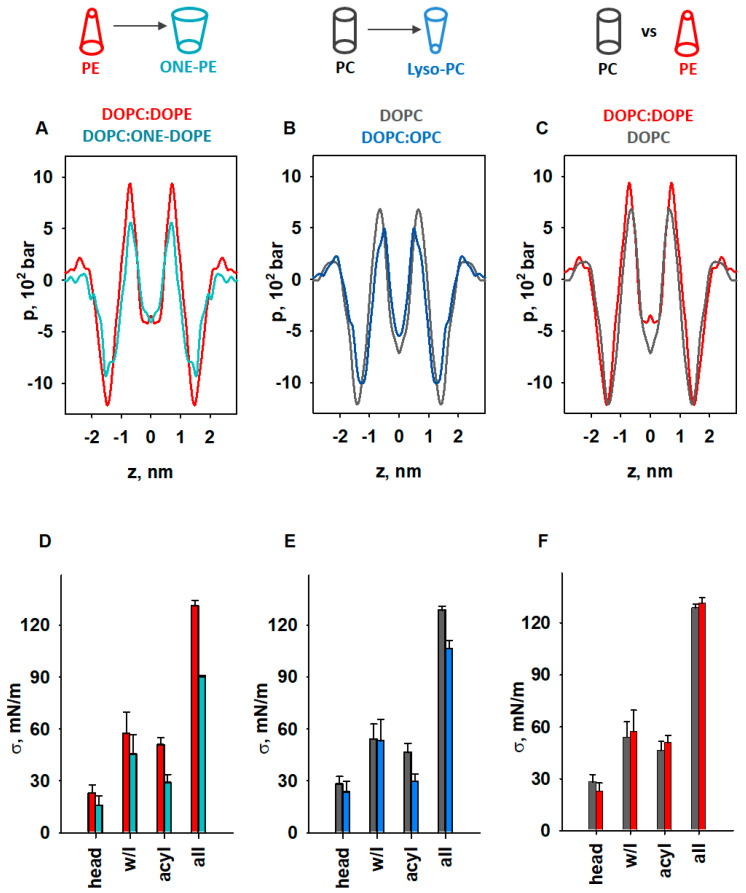
MD simulations of LPP dependency on lipid shape. (**A**–**C**) Impact of lipids with pronounced SC on the LPP, *p*. (**D**–**F**) Comparison of areas below the pressure profiles in the headgroup (head), water–lipid interface (w/l), and acyl chains (acyl) and for the whole bilayer profile (all) for *p* shown in (**A**–**C**). The lipid ratio in bi-component membranes was 50:50 mol%. Color labels: DOPC:DOPE (red), DOPC (grey), DOPC:ONE-DOPE (cyan), and DOPC:OPC (blue).

**Figure 6 antioxidants-11-02314-f006:**
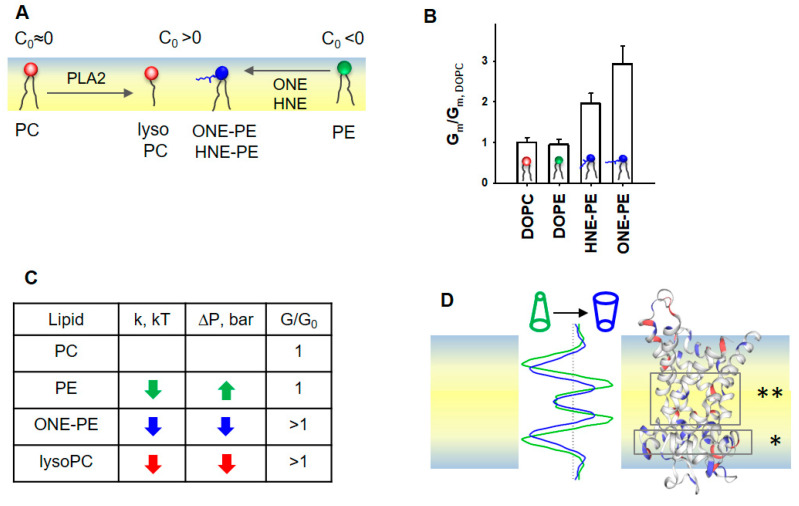
Proposed mechanism for the SCES sensing by mitochondrial proteins ANT1 and UCP1. (**A**) Schematic representation of the lipid membrane leaflet and change in lipid shapes under oxidative stress in mitochondria. (**B**) Effect of PE adducts, ONE-PE and HNE-PE, on the UCP1-mediated *G_m_* in the presence of *AA*. The relative membrane conductance *G_m_/G_m,DOPC_* was calculated based on data from [37]. *G_m,DOPC_*, and *G_m_* are specific membrane conductance in the absence or presence of PE or RA-PEs, respectively. (**C**) Schematic representation of the LPP redistribution in the lipid bilayer membrane caused by a change in the lipid shape. A comparison of the LPP (left) and protein structure (right) suggests that the decreased lateral pressure (dark blue), which appeared in the area of the FA binding site (*) and in the protein cavity region (**), promotes FA translocation. Most likely, the transformed lipid environment (i) increases the probability that anionic FA reaches a protein binding site and (ii) facilitates a protein conformational change, which in turn supports a faster translocation of FA to the opposite leaflet. (**D**) The increase in the UCP1- and ANT1- mediated *G_m_* correlates with a decrease in both the lateral pressure, *p,* and bending rigidity, *k*, in the lipid bilayer membrane. If the change in *p* and *k* goes in the opposite direction, the protein-mediated *G_m_* is not affected, as demonstrated here for the DOPE-containing membrane.

## Data Availability

The data supporting the findings are available in the Supplementary Information and from the corresponding author upon reasonable request.

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
