# Peer review of "Membrane Lipid Reshaping Underlies Oxidative Stress Sensing by the Mitochondrial Proteins UCP1 and ANT1"

_antioxidants, 2022, doi:10.3390/antiox11122314_

Round 1
Reviewer 1 Report
Please find the attached file.

Author Response
s. attached file

Reviewer 2 Report
There’s no doubt that oxidative stress-mediated damage of lipidic constituents of cellular membranes should be considered as a basis of pathogenesis of numerous diseases. Thus, deciphering molecular mechanisms that govern these pathogenic conditions is of primary interest. In their manuscript entitled “Membrane lipid reshaping underlies oxidative stress sensing by the mitochondrial proteins UCP1 and ANT1” Jovanović and collaborators describe their efforts to refine the current picture of the influence of phosphatidylethanolamine and its reactive aldehyde (RA) derivatives on mechanical properties of inner mitochondrial membranes (IMM) and some integral proteins that reside in it. The authors employed simple model membrane systems composed of up to three different lipid species together with well-established biophysical approaches and molecular dynamics (MD) simulations. This led them to some interesting observations, which could be of broad interest and potentially high influence. In general, the way the results were presented and a consequence in methodological design are major strengths of the manuscript. However, I do believe that the whole story could gain much more impact after reconsidering a few major issues listed below.
- First of all, the authors should more thoroughly discuss and justify the criteria of selection of lipids to construct the membrane systems in light of recent data on mitochondrial membrane composition. For example, Renne et al. (EMBO J 2021 41:e106837) reported the abundancy of phosphatidylethanolamine (PE) molecular species 34:2 (sn-1-palmitoleoyl-sn-2-oleoyl-phosphatidylethanolamine), and 34:1 (sn-1-palmitoyl-sn-2-oleoyl-phosphatidylethanolamine and sn-1-stearoyl-sn-2-palmitoleoyl-phosphatidylethanolamine). Since in the current manuscript 36:2 PE and PC species were used, the question arises how would shorter acyl chains and mixed saturated/unsaturated configuration affect behavior of PE? All the mentioned synthetic PE species are freely available on the market.
- In line with the previous, it is rather hard to understand that in some experiments model membrane systems containing cardiolipin (CL) were used, while in case of the others (sections 3.3 and 3.5) this lipid was omitted. Such inconsistency cannot be simply justified as it was done by the authors within l. 330-333. Firstly, it is hard to agree that 10mol% of CL is negligible as it is widely known that main lipid components of the IMM in humans are CL, phosphatidylcholine (PC) and (PE) at the approximate ratio of 1:2:2 by weight (e.g. Daum BBA 1985, 822:1). Secondly, cardiolipin-dependent properties of ternary lipid bilayers that mimic mitochondrial membranes were already studied, e.g. via MD by Wilson et al. (Biophys J 2019, 117:3) who showed significant influence on lipid diffusion and induction of negative stress.
- The authors should also refer to other studies where lateral pressure profiles of DOPC-DOPE bilayers were prepared (e.g. Orsi&Esseex Faraday Discuss 2013, 161: 249) and discuss the differences.
- It is not fully clear why the experiments were done at 32 °C, while MD simulations were performed at 310 K. Is there any particular reason behind this temperature shift? How the results could be affected by this fact?
- Were the suggested modifications of PE by reactive aldehydes (section 3.2) detected/proven during the experiment (e.g. by mass spectrometry)?
- There are only a few grammar (e.g. l.136) and typeset (e.g. l.37) errors which await correction. Also, “UCP1” on Fig. 4 should be properly aligned not to generate confusions.
- The authors should also more carefully introduce abbreviations, i.e. the parameter “SC” should be defined (how it is calculated) and explained, all abbreviations should be defined at their first use (e.g. “RAs” appear within l.66 but is defined within l. 244-245, and there are much more of such inconsistencies).
Author Response
S. attached file

Round 2
Reviewer 1 Report
The authors considered most of my comments in a good manner, but the detailed discussion I asked for regarding the comparison of in silico results with the experimental ones (Fig 5 D-E and Fig 3), discussion inserted in page 18 (rows 404-423), is not in agreement with a sentence remained from the old version of the manuscript, sentence that was the issue of my comment (in the revised manuscript page 10, row 382). Please, modify this sentence according to the discussion in page 12.
Author Response
We thank the reviewer for his/her helpful comment.
We corrected the sentence to:
"Importantly, the changes in LPP caused by the introduction of DOPE or ONE-PE or OPC into the DOPC bilayer are in qualitative agreement with the lateral stress measurements (Figs. 2 and 3)."
Reviewer 2 Report
I would like to thank the authors for addressing all the issues raised by the reviewers. The manuscript is now substantially improved and ready for publication in its current form.
Author Response
Thank you for the positive comment.